# Drive for Thinness Predicts Musculoskeletal Injuries in Division II NCAA Female Athletes

**DOI:** 10.3390/jfmk4030052

**Published:** 2019-08-01

**Authors:** Jennifer L. Scheid, Morgan E. Stefanik

**Affiliations:** 1Health Promotion Department, Daemen College, Amherst, NY 14226, USA; 2Athletic Training Department, Daemen College, Amherst, NY 14226, USA

**Keywords:** female athlete triad, college athletes, injuries, Division II, drive for thinness

## Abstract

The female athlete triad is the interrelation of low energy availability, menstrual dysfunction, and low bone mineral density. Previously, the components of the female athlete triad have been linked to bone stress injuries. The objective of this study was to explore the relationship between drive for thinness, a proxy indicator of low energy availability, and musculoskeletal injuries. Fifty-seven female athletes, from an NCAA Division II college, were followed throughout their respective sport season for musculoskeletal injuries. Women were grouped based on a median split of the drive for thinness score (high drive for thinness (DT) vs. low DT). At the end of each sport season, injury data were compiled using an electronic medical record database. Forty-seven of the 57 women (82%) incurred 90 musculoskeletal injuries. The most prevalent injuries included: Low back pain/spasm/strain (*n* = 12), followed by shin splints/medial tibial stress syndrome (*n* = 9), general knee pain (*n* = 7), quadriceps strain (*n* = 6), and knee sprain (anterior cruciate ligament, posterior cruciate ligament, medial collateral ligament, and lateral collateral ligament sprains; *n* = 5). The number of in-season injuries in the High DT group (2.0 ± 0.3) was significantly higher than the Low DT group (1.2 ± 0.2, *p* = 0.026). A high drive for thinness is associated with an increased number of injuries during the competitive season.

## 1. Introduction

Since the 1960s scientists and clinicians have been reporting menstrual irregularity in female athletes [1,2]. In 1997, the American College of Sports Medicine (ACSM) published the first Female Athlete Triad position stand, that described a syndrome of three interrelated conditions: Disordered eating, amenorrhea, and osteoporosis [3,4]. In 2007, an update on the Female Athlete Triad was published that redefined the syndrome to include: Low energy availability/disordered eating, menstrual dysfunction, and low bone mineral density [5]. The prevalence of exhibiting all three components of the Female Athlete Triad is low (1–15.9%) [6]. However, the prevalence range for presenting with one component of the female athlete triad is 16–60%, and presenting with any two components is 2.7–27% higher than individuals not at risk [6]. Research suggests that the prevalence of the female athlete triad is higher in sports that emphasize a lean physique and/or low body weight [7].

Previously, individual components of the female athlete triad have been linked to stress fractures [8] and other musculoskeletal injuries [9,10]. Rauh et al. examined the individual components of the female athlete triad (oligomenorrhea/amenorrhea, low bone mineral density, and disordered eating) and predicted the number of musculoskeletal injuries in high school-aged female athletes based on these components [9]. Rauh et al. also demonstrated that oligomenorrhea/amenorrhea (self-reported in the last year), low bone mineral density (BMD *z* score ≤ 1.0 SD), and disordered eating (defined by elevated body shape concern on an eating behavior questionnaire) all predicted the number of in-season musculoskeletal injuries [9].

In 2011, Gibbs and colleagues confirmed that a drive for thinness score, calculated via a written questionnaire, can be a proxy indicator of an energy deficiency [11]. The exercising women with higher drives for thinness scores demonstrated suppressions in resting metabolic rate and triiodothyronine indicating an energy deficiency and risk for the female athlete triad [11]. In the current study, we wanted to see if the drive for thinness score could be used to predict musculoskeletal injuries during college sports seasons.

The purpose of this study was to explore the relationship between drive for thinness and musculoskeletal injuries in college-level female athletes. We hypothesized that athletes who classify as having a higher drive for thinness would have a greater occurrence of musculoskeletal injuries during their competitive seasons.

## 2. Materials and Methods

### 2.1. Participants

The following study was conducted with the participation of female athletes from a National Collegiate Athletic Association (NCAA) (Indianapolis, Indiana) Division II college. The study included women from six of the women’s varsity sports teams: Women’s basketball, volleyball, track, cross-country, triathlon, and soccer. Women had to be at least 18 years old to participate in the study. Following a practice, all athletes interested in participating in the study gave informed consent and filled out a general health assessment and an Eating Disorder Inventory-3 (EDI-3) questionnaire [12]. Injury data were compiled for each athlete at the end of the season. The college’s institutional review board approved the study (Daemen College Human Subject Research Review Committee, E.ATH0716.164, 8 November 2016). Written informed consent was obtained from all the subjects prior to participation in the study.

### 2.2. Data Collection

#### 2.2.1. General Health Assessment

A general health assessment asked the athletes to self-report race/ethnicity, age, anthropometric measures (e.g., height and weight used to calculated body mass index), current hormonal contraceptive use, age of menarche, number of menstrual cycles in the past 12 months, history or current diagnosis of an eating disorder, and previous sport-specific injuries (in the past 5 years).

#### 2.2.2. Drive for Thinness

A Drive for Thinness score (DT) was obtained from the EDI-3 questionnaire [12]. The questionnaire asks participants to circle which applies to you (Always, Often, Sometimes, Rarely, and Never) on a number of statements. Examples of questions on the Drive for Thinness Subscale include: I eat sweets and carbohydrates without feeling nervous, I think about dieting, I feel extremely guilty after overeating, I am terrified of gaining weight, I exaggerate or magnify the importance of weight, I am preoccupied with the desire to be thinner, and If I gain a pound, I worry that I will keep gaining. Women were grouped based on a median split of their drive for thinness score (High DT (score of 2 or greater) vs. Low DT (score of less than 2)). Scores for drive for thinness can range from 0 to 21. Patients with eating disorders typically score higher (>12) [13].

#### 2.2.3. Injuries

The Athletic Trainers used an electronic medical record database (SportsWare Online; Computer Sports Medicine, Inc., V1.80.3.0. 2016, Stoughton, MA, USA,) used to manage, report, and record all pertinent athletic training information. At the end of each of the participants’ sports seasons, musculoskeletal injury data were compiled using SportsWare for all of those athletes agreeing to participate. The data collected included injury diagnosis and the total time of sport participation lost.

#### 2.2.4. Data Analysis

All continuous variables were compared using independent *t*-tests (Low DT vs. High DT). Chi-square tests were used to explore the relationship between the Low DT Group and the High DT group for any categorical variable. An ANOVA was used to compare variables of interest (the drive for thinness score and the number of injuries) among teams. Forward multivariate stepwise regression using *p* = 0.05 for entry, and *p* = 0.10 to leave in the model was used to explore predictors of the number of in-season injuries. Variables included in the first model were variables associated with the female athlete triad, including a history of stress fractures, age of menarche, history of amenorrhea, oral contraceptive use, and drive for thinness. Variables included in the second model included subscales of the eating disorder inventory including body dissatisfaction, interoceptive awareness, drive for thinness, bulimia, ineffectiveness, interpersonal distrust, perfectionism, maturity fears, and the sum score. All data were analyzed using the SPSS for Windows (version 23.0, Chicago, IL, USA) statistical software package. Data are reported in mean ± SEM.

## 3. Results

The sample included female athletes from the following teams: Women’s basketball (*n* = 11), volleyball (*n* = 14), track/cross-country/triathlon (*n* = 19), and soccer (*n* = 13). The women were 18 to 22 years of age with a BMI ranging from 18.4 to 29.6 kg/m^2^. The women were 35%, 28%, 14%, and 23% college freshman, sophomores, juniors, and seniors, respectively. Ninety-one percent of the women were Caucasian.

The median drive for thinness score was 1. Thirty-three women had a low drive for thinness based on the median split (Low DT group) and 24 women had a high drive for thinness based on the median split (High DT group). The range of drive for thinness scores was 0 to 15. Two of the athletes had a drive for thinness score >12 indicating that while they did not have a documented eating disorder, they were demonstrating eating disorder tendencies.

Table 1 shows the eating disorder inventory scores of in Low DT group compared to the high DT group. By design, the High DT group had higher drive for thinness scores (*p* < 0.001), but they also had higher body dissatisfaction scores (*p* < 0.001), perfectionism scores (*p* < 0.001), and EDI sum scores (*p* < 0.001). There were no group differences (*p* > 0.05) in the other EDI subscales.

Table 2 shows the demographic and training characteristics of the women with a low drive for thinness (DT) compared to women with a high-risk DT. The women in the Low DT group were a similar age, height, weight, BMI, and training per week compared to the High DT group (*p* > 0.05).

Forty-seven of the 57 women (82%) incurred 90 musculoskeletal injuries. The most prevalent injury was low back pain/spasm/strain (*n* = 12), followed by shin splints/medial tibial stress syndrome (*n* = 9), general knee pain (*n* = 7), quadriceps strain (*n* = 6), and knee sprain (anterior cruciate ligament, posterior cruciate ligament, medial collateral ligament, and lateral collateral ligament sprains; *n* = 5). The total number of injuries was totaled for each athlete and the women in the High DT group had more injuries compared to the Low DT group (Figure 1, *p* = 0.026). The number of days missed from injury in the Low DT group (6.9 ± 2.6 days) was not different compared to the High DT group (9.8 ± 2.6 days, *p* = 0.460).

Using pooled data for a forward stepwise linear regression, we entered variables related to the female athlete triad and subscales of the eating disorder inventory into the two prediction models respectively. For the first time in college athletes, we report that the strongest predictor of in-seasons injuries in both models is drive for thinness (adjusted R^2^ = 0.063, *p* = 0.034, Figure 2).

Since over-training is also commonly associated with injuries in sport, we created a third regression model, to control for individual hours of training per week, using the drive for thinness score and individual hours of training per week. In the regression model, only the drive for thinness score (*p* = 0.025) but not training per week (*p* = 0.116) predicted the number of injuries (adjusted R^2^ = 0.086, *p* = 0.036), indicating that drive for thinness adjusted for training hours per week still predicted the number of injuries in these athletes.

We explored if there were any team differences in the drive for thinness score or injury by sport (basketball, volleyball, track/cross-country/triathlon, and soccer). There were no differences among the teams in the drive for thinness score (*p* = 0.128). However, the volleyball team had a greater number of injuries (3.4 ± 0.9) compared to basketball (1.9 ± 1.2), track/cross-country/triathlon (1.0 ± 0.3), and soccer (2.1 ± 0.6) teams (*p* < 0.001).

## 4. Discussion

This is the first study to date, to demonstrate that female collegiate athletes with a high drive for thinness incur more musculoskeletal injuries than women with a lower drive for thinness. The women in the high drive for thinness group had a 69% increase in the number of injuries compared to the low drive for thinness group.

Previously, exercising women with a higher drive for thinness demonstrated suppressions in resting metabolic rate and triiodothyronine, indicating an energy deficiency [14]. These relationships between a high drive for thinness and suppressed resting metabolic rate and suppressed triiodothyronine concentrations were later confirmed in a larger population of exercising women [11]. Drive for thinness is a subscale of the EDI-3, and is a proxy indicator of energy deficiency [11,14] and may indicate risk for the female athlete triad.

While we examined the relationships between drive for thinness and musculoskeletal injuries in Division II college athletes, Tenforde et al. used a female athlete triad cumulative risk assessment score [15] to measure the risk of the female athlete triad in college-aged athletes, but only looked for bone stress injuries as an outcome [16]. We did not find an increase in bone stress injury, however, this study was not powered to examine only bone stress injuries; we found an increase in the number of musculoskeletal injury incidence in the women with a high drive for thinness. Rauh et al. have previously demonstrated that the components of the female athlete triad predicted the number of musculoskeletal injuries in high school-aged female athletes (track and field, cross-country running, soccer, softball, tennis, swimming, volleyball, and lacrosse) [9]. In the current study, only our whole sample of female athletes (basketball, volleyball, track/cross-country/triathlon, and soccer) demonstrated an association between drive for thinness and the number of injuries. When we explored the individual teams, we did not find any differences in the drive for thinness score, but we did demonstrate that the volleyball players had more injuries than the other teams.

Over the past 10 years, it has become commonly accepted that the root cause of the female athlete triad is low energy availability [5]. Energy availability is the amount of dietary energy available after considering exercise energy expenditure (energy in minus exercise energy expenditure) and physiological functions needed in the body [17]. Low energy availability plays a primary role in causing menstrual cycle disturbances including amenorrhea in female athletes [17,18,19]. Increasing energy availability by increasing energy intake has been demonstrated, in a case study, to help athletes with the female athlete triad by reversing the energy deficiency without compromising performance [20]. In addition to the low energy availability directly negatively impacting musculoskeletal health, hypoestrogenemia (demonstrated by some menstrual cycle disturbances) can also have an independent negative effect on musculoskeletal health [21]. Low energy availability leads to bone or tissue weakness, either directly through either low energy availability, or in combination with low estrogen. Risk of the female athlete triad could also be associated with general fatigue or overtraining syndrome that could also increase the risk of injuries in female athletes.

In the current study, we included women on hormonal contraceptives and we do not know if the women would have, or would not have, presented with a menstrual cycle disturbance if they were not taking the hormonal contraceptive. In a survey conducted by Haberland et al., 92% of physicians reported prescribing oral contraceptives to exercising women with amenorrhea [22]. While oral contraceptives have a positive effect on bone mineral density and decrease the risk of injury in female athletes [23], oral contraceptives may be masking an energy deficiency in some athletes who would otherwise have amenorrhea. We think it is important to establish a way to include female athletes on oral contraceptives when assessing risk for the female athlete triad. High percentages of female athletes (28–63%) use hormonal contraceptives [16,24,25] and menstrual manipulation using hormonal contraceptives is common among both recreationally active (69%) and competitively active (73%) female athletes [25]. In the current study, we included women on oral contraceptives and indirectly (through drive for thinness) focused on the relationship between an energy deficiency and musculoskeletal injuries. We did not find any difference in the variables of interest (drive for thinness or number of injuries) between women taking oral contraceptives and women not taking oral contraceptives.

The current study has a number of strengths. For example, this study included a variety of athletes: Women’s basketball, volleyball, track/cross-country/triathlon, and soccer, which will increase the generalizability of the study to a larger group of female athletes and not just runners (many studies on the female athlete triad and injuries focus on runners). Additionally, this study also included women on hormonal contraceptives. When researchers exclude women on oral contraceptives they eliminate a large number of athletes that may or may not be at risk for the female athlete triad. The number of female athletes taking hormonal contraceptives in a recent study conducted on college-level athletes was 28% [16]. These numbers are in line with national statistics that report one out of four women (22.4%) aged 15–24 use hormonal contraceptives [26]. However, in the current study, 63% of our female athletes were found to be taking hormonal contraceptives. Torstveit and Sundgot-Borgen investigated elite Norwegian athletes and Norwegian non-athletes and suggest that a higher percentage of elite athletes (40.2%) than non-athletes (27.5%) use oral contraceptives [24]. We encourage other studies to include women on oral contraceptives so research can be completed including this large population of the collegiate athletes.

There are limitations in the current study. While the injury data were objectively collected with the help of the athletic trainers, all of the other data used in this study were self-reported. Future studies with larger samples sizes may be able to determine which specific injuries athletes with a higher drive for thinness are at an increased risk for. We saw a large number of injuries in both the women with a low and high drive for thinness; however, the current study was not powered to determine if any specific injury was more prevalent in the High DT group compared to the Low DT group. Future research should also assess injury prevention strategies and treatment strategies among women with a high drive for thinness or women who show other risk factors of the female athlete triad.

## Figures and Tables

**Figure 1 jfmk-04-00052-f001:**
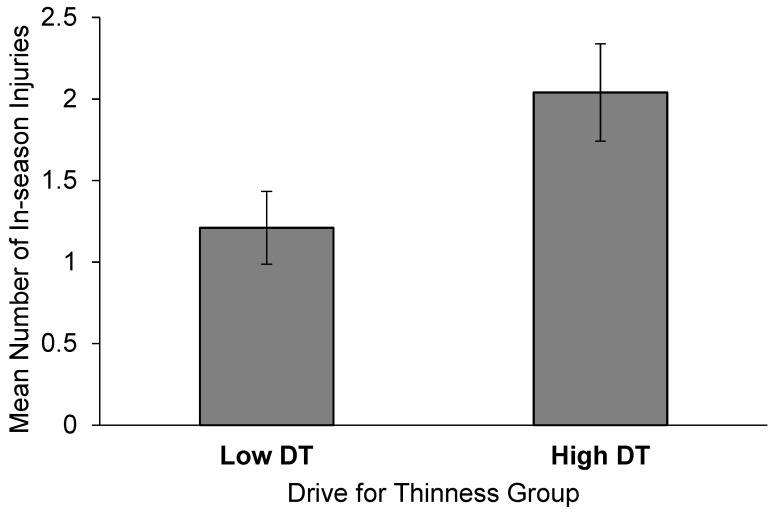
The mean number of in-season injuries in women high drive for thinness (High DT, determined by a median split) was higher (*p* < 0.05) than the women with a low drive for thinness (Low DT). Data are reported in mean ± SEM.

**Figure 2 jfmk-04-00052-f002:**
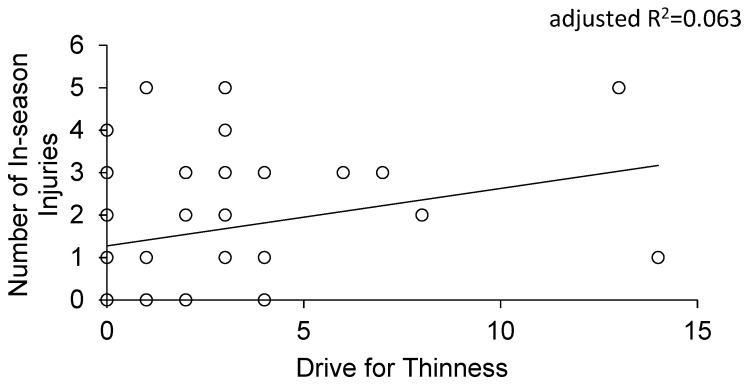
The correlation between the drive for thinness score and in-season musculoskeletal injuries in NCAA division II athletes (*p* = 0.034).

**Table 1 jfmk-04-00052-t001:** Eating Disorder Inventory (EDI) scores of the women with a low drive for thinness and high drive for thinness.

	Low DT (*n* = 33)	High DT (*n* = 24)
Body dissatisfaction	1.7 ± 0.4	5.8 ± 0.8 *
Interoceptive awareness	1.4 ± 0.4	2.0 ± 0.4
Drive for thinness	0.3 ± 0.1	4.5 ± 0.6 *
Bulimia	1.9 ± 1.2	1.6 ± 0.4
Ineffectiveness	0.9 ± 0.4	1.6 ± 0.5
Interpersonal distrust	1.6 ± 0.3	1.9 ± 0.6
Perfectionism	5.9 ± 0.5	9.6 ± 0.7 *
Maturity fears	2.9 ± 0.4	5.4 ± 0.9
Sum	14.4 ± 2.1	30.8 ± 3.8 *

* *p* < 0.05.

**Table 2 jfmk-04-00052-t002:** Demographic and the training characteristics of the women with a low drive for thinness and a high drive for thinness.

	Low DT (*n* = 33)	High DT (*n* = 24)
Age (yr)	19.0 ± 0.2	19.1 ± 0.9
Height (m)	1.65 ± 0.02	1.68 ± 0.02
Weight (kg)	60.7 ± 1.5	64.0 ± 1.8
Body Mass Index (kg/m^2^)	22.2 ± 0.3	22.6 ± 0.5
Team Training Per Week (hours)	14.0 ± 0.9	13.2 ± 1.4
Non-Team Training Per Week (hours)	2.4 ± 0.3	3.2 ± 0.2

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
