# Peer review of "Drive for Thinness Predicts Musculoskeletal Injuries in Division II NCAA Female Athletes"

_jfmk, 2019, doi:10.3390/jfmk4030052_

Round 1

Reviewer 1 Report

The manuscript os retrospective case series about the correlation between the drive for thinness(DT) and musculoskeletal injuries in college-level female athletes. Fifty-seven patients were included in the study and divided into 2 groups according to the DT score questionnaire. The results of the 2 cohorts were compared and the authors concluded that higher drive for thinness had a greater number of musculoskeletal injuries during their competitive season.

Comment 1: the sample was composed of Basketball, Volleyball, Track/Cross-country/Triathlon, and Soccer player. Could be interesting for the reader to understand the creation between the DT and sport.  Musculoskeletal injuries occurred more in a specific group of players?

Comment 2: a  superior, non-significative, individual training hours number in High DT group was found. How do the authors interpret this data? The major number of musculoskeletal injuries should be correlated to over-training?

Reviewer 2 Report

Title: Drive for Thinness Predicts Injuries in Division II 2 NCAA Female Athletes

Authors, Scheid and Stefanik has conducted an elegant study on collegiate female athletes (various sports) to predict muscular injury using EDI-2 questionnaire.  Authors have done an impressive and novel work. However, there are some issues with the manuscript which needs to be fixed before it is publishable.

Title: It is recommended that authors add "muscular injury" (to be more specific)

Abstract: no issues

Introduction:

The first sentence gives me an impression that the paper is about bias among female athletes - which is not true, it is suggested that the authors remove this sentence and be specific to athletic triad syndrome or muscular injury.

Second sentence, please clarify that the students were female and add a reference to this statement

For the following sentence: In 1997, the American College of Sports Medicine (ACSM) published …… amenorrhea, and osteoporosis [4]. - it is highly recommended that authors add following reference here and revise the sentence as needed - Curr Opin Obstet Gynecol. 2017 Oct;29(5):301-305. doi: 10.1097/GCO.0000000000000396.

Rest of the paragraph looks great!

For the following sentence: Rauh et al. demonstrated that oligomenorrhea/amenorrhea…

It is recommended that authors revise it as follows:  Rauh et al. also demonstrated that oligomenorrhea/amenorrhea….

For the following sentence: In 2011, Gibbs and colleagues confirmed that drive for thinness score, a paper and pencil questionnaire, can be a proxy indicator of an energy deficiency [11].

It is recommended that the authors consider revising this sentence.  It can be read as follows: In 2011, Gibbs and colleagues confirmed that the drive for thinness score, calculated via a written questionnaire, can be a predictor of an energy deficiency [11].

The next sentence on RMR is strong and needs a reference.

For the next sentence: In the current study, we wanted to see if a paper and pencil questionnaire, drive for thinness score, could be used to predict musculoskeletal injuries during college sports seasons.

Currently the statement is emphasizing on pencil and paper rather than the importance of survey itself as a tool.

It is highly recommended that the authors revise it as follows:  In the current study, we wanted to see if drive for thinness score could be used to predict musculoskeletal injuries during college sports seasons.

Materials and methods

Six types of athletes were included - were there any differences between them? This would be interesting for other audiences.  Can be included in supplementary materials.

For the last sentence: Consent was document by all athletes prior to participation of the study.

It is highly recommended that authors revise this sentence as follows: Written informed consent was signed/obtained by/from all the subjects prior to participation of the study.

For section 2.2.2 - it is recommended that the authors add the range of score for EDI-2 questionnaire, i.e., min-max score, at the end of the paragraph.

Results

The BMI of subjects is 15.8-29.6 kg/m^2. It looks like some of the athletes had ED? Authors need to discuss this in the discussion. 

The proportion of junior, sophomore, junior, and senior looks well distributed.  Did authors explore the number of injuries in each group? This could potentially tell the role of experience in sport and injury? - can be added in the supplementary if any differences are found.

The second paragraph, 1st and 2nd sentences (Women were grouped according……was used to determine the groups.) belongs in the methods section.

Figure 1. No description of error bars - are these SD or SEM? Authors are suggested to clarify this in the figure 1's legend.

Last paragraph - line 138.  For the readers' interest, it is highly recommended that the authors add a graph for this relationship.

Discussion

It is highly recommended to revise first paragraph. First sentence is a great start but the next sentence is weak. It's almost better if the authors can remove it. Currently, it emphasizes on use of paper and pencil and the use of questionnaire rather than the relationship between injury and drive for thinness.

For the following sentences:  Tenforde et al. used a female athlete triad cumulative risk assessment….previous stress fractures were predictors of bone stress injuries.

Too much time is spent on this topic.  Why discuss if this is not related to current study? It is highly recommended that the authors only discuss the studies that are related to the present study. Were there no injuries in the present stud that were related to bone stress?

For the following sentence:  ……musculoskeletal injuries in high school aged female athletes (track and field, cross-country running, soccer, softball, tennis, swimming, volleyball, and lacrosse) [9].

It is highly recommended that authors add a statement describing (compare/contrast) results from present study after this sentence.

For the following sentences: Over the past 10 years, it has become commonly accepted….energy availability plays a primary role in 163 causing menstrual cycle disturbances including amenorrhea in female athletes [14–16].

It is recommended for authors to add a following statement or a better statement at the end of this chain of sentences:  These effects were recovered by replenishing energy intake (Sports 2018, 6(3), 82; https://doi.org/10.3390/sports6030082)

For the following sentences:  In the current study, we included women on hormonal contraceptives…..were not taking the hormonal contraceptive.

Please refer to following citations and revise these sentences as needed.

Int J Sports Physiol Perform. 2018 Jan 1;13(1):82-87. doi: 10.1123/ijspp.2016-0689. Epub 2018 Jan 23.

Sports Med. 2017 May;47(5):869-886. doi: 10.1007/s40279-016-0636-4.

For the following sentences: The most concerning impact of oral…..impacted first-pass effect can lead to a decrease in bone formation.

Not sure if IGF mechanism needs to be discussed.  It is recommended that authors discuss only mechanisms that are specific to findings of the present study.

Strengths looks great!

Were there any differences between type of sport and number of injuries?

Since 63% of the subjects in the present study were OC users, were there any differences between OC vs non-OC users?

For the following sentences: Another strength of the current study is the simple, low technology way….participation exam to determine drive for thinness.

Not sure how this is a strength of this study. Are authors trying to talk about EDI-2? It is hard to understand what authors are trying to say. It is highly recommended to remove or revise these sentences so that it is clear and understandable.

Sample size is not a limitation

For the following statement: … we did not have anyone in the study with a current eating disorder….

Not sure how this is true when there are subjects with BMI of 15.8%. Authors need to address this.

The last second paragraph is strong and well written - this could be the ending for the manuscript.  The last paragraph is almost not needed.  If authors wish to include last paragraph in the manuscript, it is highly recommended to switch last two paragraphs and make the future studies paragraphs as the last paragraph.

Authors contributions needs to be disclosed

Funding information is missing

Reviewer 3 Report

Overall this an interesting topic that fits the journal well and is interesting. However, the paper is not an experimental design and any type of inferences should be used with caution. For example, the final sentence in the abstract should be softened as High DT does not increase the risk specifically but there may be underlying factors associated with the high DT group that leads to additional seasonal injuries. Also to simple assume a drive for thinness extrapolates to low energy availability is far reaching. It is not appropriate to assume such a link without corresponding evidence to support it. I also caution extending such conclusions when all height and weight measurements (therefore BMI) were subjective and self-reported.

Abstract

Please reword final sentence to summarize.

Introduction

The first few sentences do not support the title on content of the study as it pertains to perceived thinness or body image and composition as it relates to injuries in female athletes. I recommend editing so the intro paragraph provides a good indicator about the overall impact of the study.

Methods

Why was body composition not directly measured? For this type of study, at least height and weight should be objectively measured and not self-reported, especially with females and the topic relates to perceive thinness and body image. I believe this is a rather significant methodological flaw.

Results

More information about the type of injury sustained is needed. You state the following, but there does not appear to be any analysis about the severity of injury and thinness group. This information would make for a more impactful paper. “At the end of each the participants sports seasons, musculoskeletal injury data was 87 compiled using Sportsware for all of those athletes agreeing to participate. The data collected included date of injury, mechanism of injury, injury diagnosis, limb involved, and total time of sport 89 participation lost.”

Where is the energy availability data? Simply saying drive for thinness is a proxy for low energy availability is not scientifically sound. Do you have some other types of data (surveys) to support low energy availability?

Discussion

The first sentence needs to be supported with the results…please add percent’s of valuable information to help the reader understand the impact of the High vs low DT.

You talk about energy availability, but were in your results do you provide details or assess energy availability? This appears to be a main idea in the paper with no data to support it?

Limitations needs to reflect the self-reported measures of body composition and how that can be vastly different than actual weight/body comp.

Last paragraph…what type of associate…positive or negatiave?

Round 2

Reviewer 2 Report

Title: Drive for Thinness Predicts Injuries in Division II 2 NCAA Female Athletes

Authors, Scheid and Stefanik has significantly revised the manuscript and has improved significantly.

There are minor changes that authors may want to make to improve readability of the manuscript.

Line 95-96, this is interesting.

Line 118, under methods, 2.2.4, it is recommended to clarify on what variables were used when ANOVA was conducted.  It is recommended that authors add all variables in parentheses after "variables" on line 118 or at the end of sentence on line 119.

Line 165, it is recommended that authors change "mean±SEM" to "data are reported in mean±SEM."

Line 169, it is recommended that authors delete the extra parentheses and include (adjusted R2, p, figure 2) within one parentheses.

Figure 2, it is recommended that the authors add the "p" and "R2" values on the top right corner of this figure.

From the figure 2, it does looks like 2 people have DT score >12 (eating disorder).

Line 178, adjusted for what? Please clarify.

Line 180, this is good!

Line 193, the following sentence: "subscale is a subscale of…"  This reads weird. It is recommended that authors revise this statement.

Rest of the paragraph reads well.

Line 198, it is recommended that authors delete the word "While," and replace the word "this" with "however, the present."  The sentence should read as follows from line 198-199: "bone stress injuries as an outcome [16]. We did not find an increase in bone stress injury, however, the present study was…"

Line 206, it is recommended that authors replace the word "exposed" with "find."

Rest of the paragraph is really good!

Line 259, please add the statement about no difference between OC and non-OC users in the present study at end of the paragraph.

Line 274-275, the word "include" on line 174 and the words "did not have anyone" is conflicting. This does not clarify if the protocol of this study was to exclude participants with clinical DT or did the authors not had any participants with clinical DT? It is highly recommended that authors clarify this here. However, based on graph 2, two participants appear to have DT>12, does this suggest clinical DT? It is recommended that authors make careful statements here.

Line 291, was there any internal funding? Like departmental or university internal funding. If yes, please indicate the source. If not, it is recommended that authors indicate "none."

Line 521, please bold the year "2014."

Reviewer 3 Report

All corrects were made...great job!

Author Response

Thank you for your review.